# Pathogenesis-Related 1 (PR1) Protein Family Genes Involved in Sugarcane Responses to *Ustilago scitaminea* Stress

**DOI:** 10.3390/ijms25126463

**Published:** 2024-06-12

**Authors:** Talha Javed, Wenzhi Wang, Tingting Sun, Linbo Shen, Xiaoyan Feng, Jiayan Huang, Shuzhen Zhang

**Affiliations:** 1National Key Laboratory for Tropical Crop Breeding, Institute of Tropical Bioscience and Biotechnology, Chinese Academy of Tropical Agricultural Sciences, Haikou 571101, China; talhajaved@itbb.org.cn (T.J.); wangwenzhi@itbb.org.cn (W.W.); suntingting@itbb.org.cn (T.S.); shenlinbo@itbb.org.cn (L.S.); fengxiaoyan@itbb.org.cn (X.F.); 2Sanya Research Institute, Chinese Academy of Tropical Agricultural Sciences, Sanya 572024, China; 3School of Tropical Agriculture and Forestry, Sanya Nanfan Research Institute, Hainan University, Sanya 572025, China; hjy1314hh@163.com

**Keywords:** sugarcane, PR1 proteins, biotic stress, transcript expression, defense response

## Abstract

Plant resistance against biotic stressors is significantly influenced by pathogenesis-related 1 (PR1) proteins. This study examines the systematic identification and characterization of PR1 family genes in sugarcane (*Saccharum spontaneum* Np-X) and the transcript expression of selected genes in two sugarcane cultivars (ROC22 and Zhongtang3) in response to *Ustilago scitaminea* pathogen infection. A total of 18 *SsnpPR1* genes were identified at the whole-genome level and further categorized into four groups. Notably, tandem and segmental duplication occurrences were detected in one and five *SsnpPR1* gene pairs, respectively. The *SsnpPR1* genes exhibited diverse physio-chemical attributes and variations in introns/exons and conserved motifs. Notably, four SsnpPR1 (SsnpPR1.02/05/09/19) proteins displayed a strong protein–protein interaction network. The transcript expression of three *SsnpPR1* (*SsnpPR1.04/06/09*) genes was upregulated by 1.2–2.6 folds in the resistant cultivar (Zhongtang3) but downregulated in the susceptible cultivar (ROC22) across different time points as compared to the control in response to pathogen infection. Additionally, *SsnpPR1.11* was specifically upregulated by 1.2–3.5 folds at 24–72 h post inoculation (hpi) in ROC22, suggesting that this gene may play an important negative regulatory role in defense responses to pathogen infection. The genetic improvement of sugarcane can be facilitated by our results, which also establish the basis for additional functional characterization of *SsnpPR1* genes in response to pathogenic stress.

## 1. Introduction

Climate change powers the risk of various environmental stressors worldwide and has also affected the incidence and geographic distribution of plant pathogenic diseases [1]. Pattern-recognition receptors (PRRSs) allow plants to identify the first layer of conserved pathogen-associated molecular patterns (PAMPs), which they then use to induce pattern-triggered immunity (PTI) in response to a variety of stressors [2,3]. Moreover, plants selectively detect effectors using polymorphic nucleotide-binding domain and leucine-rich repeat (NB-LRR) proteins expressed by the majority of resistance genes in order to initiate effector-triggered immunity (ETI) [2,3,4].

Pathogenesis-related (PR) proteins have been revealed to be essential for plant disease resistance in numerous studies [5,6]. Upon pathogen-induced stress perception, various signaling cascades are activated, resulting in the activation of various genes such as WRKYs, which activate the transcription of PR proteins such as PR1 by binding to their promoter region and regulate defense responses by inhibiting the growth of pathogens [3,7]. Among these, pathogenesis related-1 (PR-1) proteins induced by pathogens or salicylic acid constitute the main family of PR proteins [5,7,8]. PR-1s comprise proteins that have a highly conserved structure [5,8]. The nuclear magnetic resonance (NMR) structure of *Solanum lycopersicum* P14a (PR-1b) showed that PR-1 has a distinct α-β-α sandwich structure that has four β-folds (A-D) and four highly conserved α-helices (I-IV) [5,9]. Recently, Han and colleagues reported that the last 11 C-terminal amino acids of the PR-1 protein contain a peptide with a caveolin-binding motif and CAP-derived peptide (CAPE) [10]. CAPE-mediated signaling facilitates defense responses against pathogen attack, but other signaling cascades associated with stress, such as those elicited by PAMPs, do not influence the CAPE response [7,9,10]. The conserved CNYx motif and tablysin-15 fatty acid-binding residues, as well as a CAP tetrad and caveolin-binding motif, are other additional functional motifs found within PR-1 proteins. The caveolin-binding motif and CAP-derived peptide (CAPE) are present in the conserved CAP domain of the PR-1 protein [10]. Previously, PR-1 family members have been systematically identified and characterized in different plant species, such as 10 in *Mangifera indica* [11], 12 in *Triticum durum* [12], 20 in *Hordeum vulgare* [8], and 86 in *Triticum aestivum* [13]. A growing body of research has shown that PR1 proteins are essential for plants’ defense against biotic stressors [7]. For instance, the ectopic expression of *Mangifera indica MiPR1A* in transgenic *Arabidopsis* and overexpression of *Triticum aestivum TaPR1-7* and *Solanum lycopersicum SlPR1* attributed to enhanced disease resistance in *Colletotrichum gloeosporioides*, *Puccinia striiformis* f. sp. *Tritici*, and *Ralstonia solanacearum*, respectively [11,13,14].

Sugarcane (*Saccharum* spp.) is a major dual-purpose cash and biofuel crop worldwide [15]. The highly polyploid and aneuploid modern sugarcane cultivars (*Saccharum* spp. Hybrid, 2n = 100–130) are derived from interspecific hybridization between *S. officinarum* (2n = 8x = 80) and *S. spontaneum* (2n = 4x–16x = 40–128) [16,17]. Additionally, the increased variation in *S. spontaneum* genome polyploidy levels contributes to the study of the evolution of autopolyploid genomes in plants [17,18]. Importantly, the two reference *S. spontaneum* clones Np-X (2n = 4x = 40, x = 10) and AP85-441 (1n = 4x = 32, x = 8) have tetraploid genomes that offer a valuable resource to uncover numerous gene families with functional diversity [17,19]. Previously, PR1 family genes were identified in the autopolyploid *S. spontaneum* AP85-441, and their temporal transcript patterns were examined in response to different stressors [5]. However, no reports are available about PR1 family genes’ responses to *U. scitaminea* infection in sugarcane. Therefore, this study presents the systematic identification and characterization of PR1 family members in sugarcane (*S. spontaneum* clone Np-X), with expression profiling in two cultivars inoculated with pathogenic fungi. This study provides a solid foundation for subsequent functional characterization (reverse/forward genetics) of PR1 family members in sugarcane in response to biotic stressors.

## 2. Results

### 2.1. Identification, Physio-Chemical Characterization, and Phylogeny of SsnpPR1 Genes

*SsPR1* genes from *S. spontaneum* clone AP85-441 were used as query sequences to search for a total of 18 *SsnpPR1* genes in the *S. spontaneum* Np-X genome by using Tbtools. The finalized *SsnpPR1* genes were named based on their corresponding sequence nomenclature from *S. spontaneum* AP85-441. Notably, the putative SsnpPR1 proteins contained 155 to 291 amino acids, molecular weights ranged from 16428 to 31117 kDa, isoelectric points ranged from 6.8 to 10.1, instability indexes fell between 40.1 and 54.7, and aliphatic indexes ranged from 59.2 to 66.2. Among all candidate proteins, the GRAVY value for SsnpPR1.13 (0.02) was positive, suggesting the hydrophobic nature of this protein (Table 1). A phylogenetic tree was constructed to better understand the evolutionary relationship among *SsPR1* genes of *S. spontaneum* clones AP85-441 and Np-X. Based on phylogeny, all *PR1* genes were clustered into four groups. Group A comprised 11 and 12 *SsPR1* and *SsnpPR1* genes, respectively. Additionally, two different genes from each clone were clustered in groups B and D separately. Group C comprised three (*SsPR1.11/15/16*) AP85-441 genes and three genes (*SsnpPR1.11/15/16*) from Np-X (Figure 1).

### 2.2. Gene Structure and cis-Regulatory Elements of SsnpPR1 Genes

Among the 18 *SsnpPR1s*, the gene structure analysis suggested that only one intron was present in four *SsnpPR1s* (*SsnpPR1.07/08/09/11*), whereas the other 14 *SsnpPR1s* lack introns. The *SsnpPR1s* contained one (*SsnpPR1.01/02/03/04/05/06*) to two (*SsnpPR1.07/08/09/11*) exons. Moreover, *SsnpPR1-7* displayed the longest intron, followed by *SsnpPR1-8* and *SsnpPR1-9*. Notably, *SsnpPR1s* had between four (*SsnpPR1.03/04/09/14/19*) and seven (*SsnpPR1.07/08*) conserved motifs. Interestingly, motifs 1 and 3 were present in all *SsnpPR1s* (Figure 2). Detailed information about a total of five groups (light-, hormone-, stress response-, meristem-, metabolism-, circadian-, and seed-related) of *cis*-regulatory elements is given in Figure 3. The highest number of *cis*-regulatory elements was found in the stress-responsive group of *SsnpPR1s*.

### 2.3. Chromosomal Mapping, Gene Duplication, and Collinearity Analysis of SsnpPR1 Genes

Chromosomal distribution revealed that highest number (six) of *SsnpPR1s* (*SsnpPR1.04/05/07/08/09/14*) were mapped on chromosome 2C. Moreover, chromosomes 1C (*SsnpPR1.11*), 2A (*SsnpPR1.12*), 2B (*SsnpPR1.13*), and 4A (*SsnpPR1.19*) had only one gene (Appendix A). Interestingly, one gene pair (*SsnpPR1.15* and *SsnpPR1.16*) displayed tandem duplication, while five gene pairs were segmentally duplicated (Figure 4). Collinearity analysis revealed the evolutionary relationship among PR1 family genes of *S. spontaneum* clones AP85-441 and Np-X. For example, *SsnpPR1.13* displayed a syntenic relationship with one *SsPR1* (Sspon.02G0020390) gene from AP85-441. Interestingly, *SsnpPR1.05* displayed syntenic associations with two different genes (Sspon.02G0024520 and Sspon.02G0024530) from AP85-441 (Figure 5). Moreover, *SsnpPR1.02/09/14/15/18* also had a syntenic association with one different gene from AP85-441. Detailed information about syntenic association is given in Table 2.

### 2.4. Protein–Protein Interaction Analysis

A strong protein–protein interaction network was predicted among four SsnpPR1s (SsnpPR1.02/05/09/19) based on their orthologs in *A. thaliana*. Additionally, SsnpPR1.19 was the key protein in this interaction network. Interestingly, no point-to-point interaction was observed among SsnpPR1s (Figure 6; Appendix A).

### 2.5. Sugarcane Cultivar Responses to Pathogen Infection

Based on qPCR data, a significantly high population size of *U. scitaminea* pathogen was recorded in ROC22 at time points as compared to that in Zhontang3. The highest population size of the pathogen in ROC22 and Zhontang3 was 1470.1 copies/µL and 961.1 copies/µL, respectively (Figure 7).

### 2.6. Transcript Profiling of SsnpPR1 Genes in Response to Pathogen Infection

The characterization of *SsnpPR1* genes responding to smut pathogen was determined by RT-qPCR assay. Overall, the results revealed four different kind of transcript profiling: (1) three *SsnpPR1* (*SsnpPR1.03/05/19*) genes were upregulated across different time points as compared to the control in two sugarcane cultivars; (2) one *SsnpPR1* (*SsnpPR1.02*) gene was downregulated across different time points as compared to the control in two sugarcane cultivars; (3) two *SsnpPR1* (*SsnpPR1.07/08*) genes displayed irregular transcript profiles; and (4) three *SsnpPR1* (*SsnpPR1.04/06/09*) genes were specifically upregulated in sugarcane cultivar Zhongtang3 but downregulated in ROC22 across different time points as compared to the control. In contrast, *SsnpPR1.11* was specifically upregulated in sugarcane cultivar ROC22 and downregulated in Zhongtang3 (Figure 8).

Specifically, the transcript expression of three *SsnpPR1* (*SsnpPR1.04/06/09*) genes was upregulated by 1.2–2.6 folds in the resistant cultivar (Zhongtang3) and downregulated in the susceptible cultivar (ROC22) across different time points as compared to control in response to pathogen infection. Additionally, *SsnpPR1.11* was specifically upregulated by 1.2–3.5 folds at 24–72 h post inoculation (hpi) in ROC22, suggesting that this gene may play an important negative regulatory role in defense responses to pathogen infection.

## 3. Discussion

### 3.1. Genetic Diversity of SsnpPR1 Genes

The PR1 gene family is highly conserved, diverse, and versatile, and the number of members varies from species to species, with significant diversity in structure and function [7]. The number of *SsnpPR1* (18) family members identified in this study was lower than that in *Hordeum vulgare* (20) [8], *Glycine max* (24) [20], *Solanum tuberosum* (22) [21], and *Triticum aestivum* (86) [13]. In contrast, the number of *SsnpPR1s* was higher than that in *Piper nigrum* (11) [22], *Triticum durum* (12) [12], *Musa acuminate* (15) [23], *Allium sativum* (16) [24], and *Camellia sinensis* (17) [25]. The possible reason behind the expansion of PR1 family members might be gene duplication events, which include two major driving forces (tandem and segmental) for the duplication of gene families in plants, particularly in sugarcane [5]. Notably, the results of the current study are in line with those of recent studies, which demonstrated that gene duplication for acquiring new genes in this family genes was a common occurrence in sugarcane [26,27,28]. Due to the complex polyploid nature of sugarcane, functional divergence and redundancy is more frequent, and the unique retention profile of duplicated genes after the establishment of polyploidy might explain a general increase in biological complexity [29]. In addition to duplication events, the influence of polyploidy on mutation rates might also cause insertion–deletion events [30].

### 3.2. Functional Divergence of SsnpPR1 Genes

Biotic and abiotic stressors have long hindered the normal functioning of plant processes and ultimately limited crop production [5,9,10]. Multiple factors are involved in stress perception, signal transduction, signaling crosstalk, gene activation, and transcription during plant–pathogen interactions that contribute to disease resistance. In recent years, significant progress has been made in understanding and discovering the broader biological roles of PR-1 family members in plant defense. A growing body of research shows that PR-1s are essential for plant defenses against pathogen attacks. The range of regulatory roles in responses to pathogen infection is likely associated with the functional diversity seen among *PR-1* genes [5,7,9,10]. For instance, different PR-1 alleles have different regulatory functions against diverse stressors [7]. Specifically, PR1 proteins play crucial roles in plant resistance and defense response against pathogens [7,31]. Notably, several tested *SsnpPR1* genes in this study had a positive regulatory role in the defense response against the pathogen in the resistant sugarcane cultivar. Furthermore, the *SsnpPR1.02/05/09/19* genes may be essential to the SsnpPR1 protein interaction network and may facilitate the critical link in sugarcane crop defense responses to smut pathogen infection. A number of significant findings from earlier research have indicated that PR-1 functions as a defense protein during plant–pathogen interactions [32,33,34]. For example, sugarcane *ScPR1* overexpression in transgenic lines was found to be involved in regulating disease resistance against *Acidovorax avenae* subsp. *Avenae* [5]. Previously, Guo and colleagues demonstrated the dynamic modulation of *MePR1* activity in defense response to *Xanthomonas axonopodis* pv. *Manihotis* in *Manihot esculenta* [35]. Interestingly, previous findings also revealed the involvement of TaPR1 protein in defense response against *Puccinia triticina* by interacting with TaTLP1 [36]. Perhaps diverse regulatory roles in response to pathogen infection might be due to the functional diversity that exists among *PR1* genes [31]. Importantly, the results of the current study provide a basis for the further exploration of mechanisms associated with signaling cascades mediated by *SsnpPR1* genes responding to pathogen infection.

CAPE peptide-mediated defense responses could also function as a trade-off between biotic and abiotic stressors. Results from Chien and colleagues support this possibility by demonstrating that the silencing of PROAtCAPE1 enhanced tolerance to salt stress. Conversely, the same transgenic plants with exogenous application of AtCAPE1 exhibited reduced survival when grown on high-salt media, and this inhibited growth could be due to the downregulated expression of genes associated with salt tolerance [37]. Notably, in wheat, the physical interaction of *Triticum aestivum* TaPR1 with TaTLP1 via the αIV helix is involved in resistance to the fungus *Puccinia triticina* (the causative agent of wheat rust) through the C-terminal CAPE1 motif [36]. PR-1 also undergoes partial proteolytic processing as it transits through multi-vesicular bodies, and this proteolysis depends on the presence of an intact C-terminal motif. However, only process-mimicking or non-mutated variants of PR-1 are extravagated to apoplasts. The proteolytic cleavage of the PR-1 C terminus releases protein fragments that modulate the defense responses of plants, particularly modulating localized cell death. Furthermore, regions of PR-1 that remain after processing may have potential roles outside of cell death, such as in host–plant ontogenesis and tissue-localized dependent activities.

## 4. Materials and Methods

### 4.1. Identification and Physio-Chemical Characterization of PR1 Family Members

To identify PR1 family members in the genome of *S. spontaneum* clone Np-X, SsPR1 query sequences from *S. spontaneum* clone AP85-441 were used [5]. Based on the similarity index (≥80%) and e-value (0) candidate sequence domain checking was performed using Pfam (http://xfam.org/; accessed on 2 August 2023), NCBI (http://blast.ncbi.nlm.nih.gov/; accessed on 2 August 2023), and SMART (http://smart.emblheidelberg.de/; accessed on 2 August 2023) domain analysis software. Additionally, the nomenclature of Chu et al. [5] was followed for the PR1 family genes found in the Np-X genome. The online ProtParam tool (https://web.expasy.org/protparam/; accessed on 5 September 2023) was used to elucidate physio-chemical properties such as number of amino acids (aas), molecular weight (Da), isoelectric point (pI), instability index, aliphatic index, and hydrophilicity of PR1 proteins.

### 4.2. Phylogenetic and Gene Structure Analysis

The candidate SsnpPR1 sequences were subjected to MEGA 7.0 software (neighbor-joining method and 1000 replicates) to construct the phylogenic tree. The EvoIView (https://www.evolgenius.info/; accessed on 5 September 2023) tool was used to beautify the phylogenetic tree. The gene structure was analyzed by GSDS (http://gsds.gao-lab.org/; accessed on 5 September 2023) and MEME Suite (https://meme-suite.org; accessed on 5 September 2023) servers with default parameters and visualized by using Tbtools v0.6655 [38].

### 4.3. Chromosomal Localization, Gene Duplication, and Collinearity Analysis

The SsnpPR1 family members were mapped on different chromosomes by using Tbtools v0.6655. Moreover, gene duplication and collinearity analysis was also performed by using Tbtools v0.6655.

### 4.4. Protein Interaction Network and cis-Regulatory Elements

The String database (http://www.string-db.org; accessed on 7 September 2023) was used to predict the protein–protein interactions among SsnpPR1 family members and their orthologs in *Arabidopsis thaliana*. The *cis*-regulatory elements were analyzed from the promoter region (1.5 kb) of each gene by using the PlantCARE database (http://www.dna.affrc.go.jp/PLACE/; accessed on 7 September 2023).

### 4.5. Crop Husbandry, Pathogen Inoculation, and Leaf Sampling

Two sugarcane cultivars, ROC22 (susceptible) and Zhongtang3 (resistant), were procured from the Institute of Tropical Bioscience and Biotechnology, Sanya Research Station, CATAS/Hainan Yazhou Bay Seed Laboratory, Sanya, Hainan (56.0411° N, 12.7009° E), for this study. Singled-budded sugarcane setts were prepared and soaked in water overnight, followed by drying for two hours at room temperature. The bud injection method was used for sugarcane plant inoculation with the smut pathogen (*U. scitaminea*). Both cultivar samples were collected at 0, 24, 48, and 72 h post inoculation (hpi).

### 4.6. Determination of Fungal Population Size by qPCR Assay

The population size of *U. scitaminea* in two sugarcane cultivars was determined by using TaqMan quantitative PCR assay according to instructions given by Su and colleagues [39].

### 4.7. Expression Profiling of SsnpPR1s by RT-qPCR Assay

RNA from emerging bud samples was extracted by using megazol reagent (Omega Bio-tek, Inc., Norcross, Georgia) according to the manufacturer’s instructions. The complete list of primers along with their sequences is given in Table 3. RT-qPCR was carried out using Analytike Jena (qTower3G), Germany. The reaction mixture comprised 10 μL of 2 × SYBR PremixEx TaqTM (Takara, Kusatsu, Japan), 2 μL of each forward and reverse primer, 1 μL of cDNA, and 6.6 μL of ddH_2_O to make a final volume of 20 μL. For each sample, three biological and technical replicates were employed. The reference gene was glyceraldehyde 3-phosphate dehydrogenase (GAPDH), and the expression of each gene was measured using the 2^−ΔΔCt^ technique.

### 4.8. Statistical Analysis

Using the statistical software program Statistix 8.1, the means of various time periods were compared through the least significant difference (LSD) test at a 5% probability level (*p* < 0.05).

## 5. Conclusions

A total of 18 *SsnpPR1* genes from the *Saccharum spontaneum* Np-X genome were identified and classified into four groups. Gene duplication via tandem and segmental events played a crucial role in the expansion and provided valuable information about the *SsnpPR1* genes’ evolutionary history. Based on the functional characterization of *SsnpPR1* genes as determined by RT-qPCR assay, we proposed that *SsnpPR1.04/06/09* may play a positive regulatory role, whereas *SsnpPR1.11* may function as a negative regulator in the defense response to smut pathogen. This study also provides a basis for the elucidation of detailed molecular mechanisms in sugarcane in response to pathogen stress using modern biotechnologies such as forward/reverse genetics. Many unanswered questions remain that will need to be addressed by future investigations. These questions include: (i) what signaling cascades are affected by PR1s; (ii) whether PR-1s have direct or indirect inhibition ability toward pathogen growth; (iii) what regulatory networks and/or signaling cascades act upstream of PR1s; and (iv) if functionally similar and/or diverse PR1s can be integrated for the development of resistant transgenic sugarcane cultivars.

## Figures and Tables

**Figure 1 ijms-25-06463-f001:**
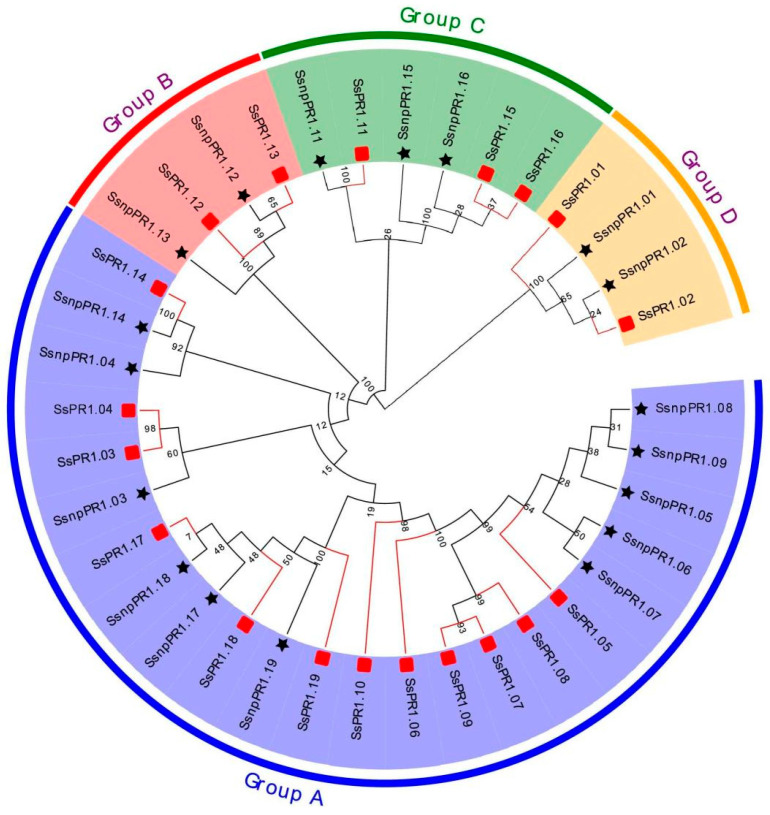
Phylogenetic tree of PR1s from *Saccharum spontaneum* clones AP85-441 and Np-X based on the neighbor-joining method with 1000 replicates. Red boxes and black stars represent *SsPR1* and *SsnpPR1* genes, respectively. Group names are depicted by different colors on the outer ring of the tree.

**Figure 2 ijms-25-06463-f002:**
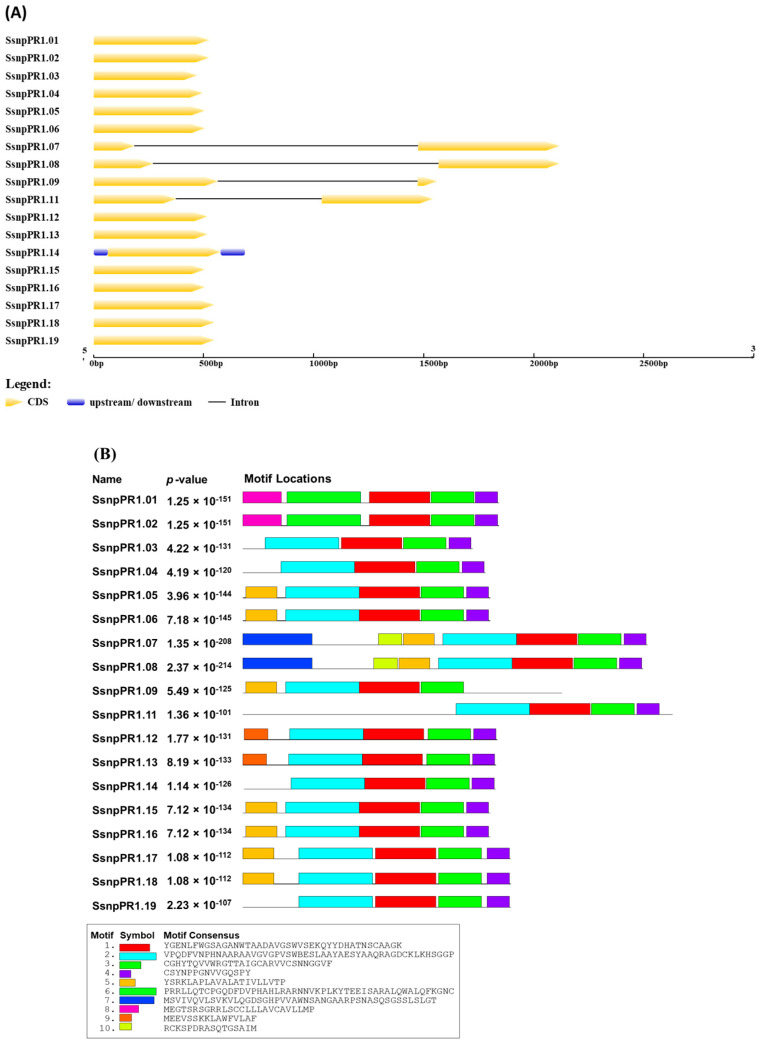
Gene structure and conserved motif locations in *SsnpPR1s*. (**A**) Intron/exon locations and lengths are shown by gray lines and yellow boxes, respectively. (**B**) Conserved motifs are shown by different colored boxes.

**Figure 3 ijms-25-06463-f003:**
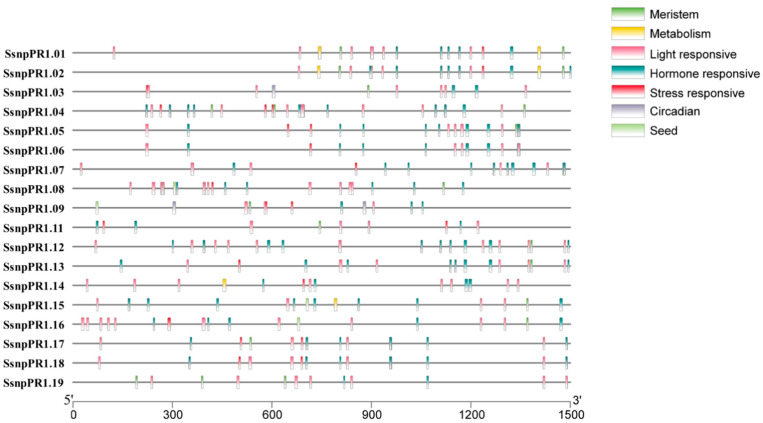
*Cis*-regulatory elements in the promoter region (1.5 Kb) of *SsnpPR1s*.

**Figure 4 ijms-25-06463-f004:**
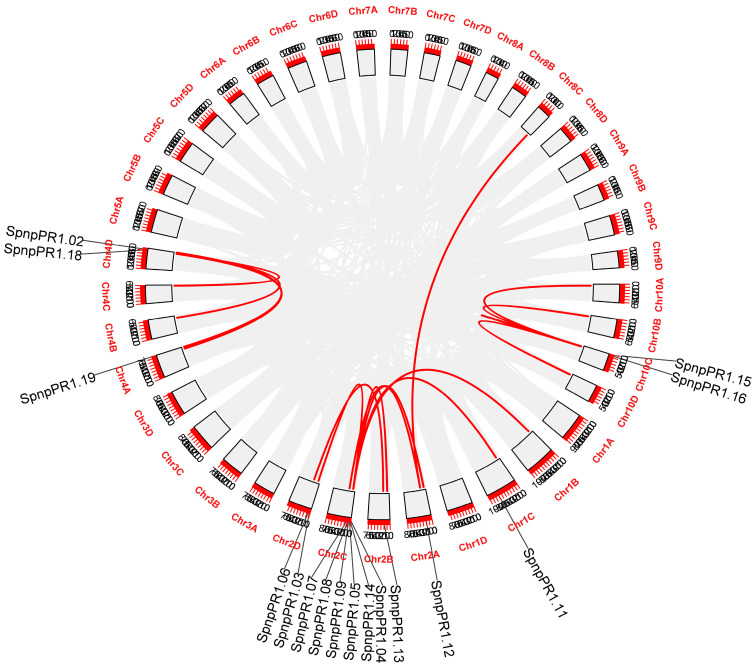
Gene duplication occurrences between *SsnpPR1* genes as indicated by red lines in the inner circle. Chromosome numbers are indicated by the red color outside of gray boxes. Gene names are present on the outer circle.

**Figure 5 ijms-25-06463-f005:**
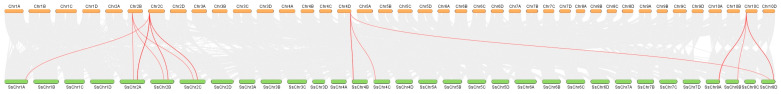
Collinearity association among *PR1s* from *S. spontaneum* clones Np-X and AP85-441. Collinearity is shown by red lines. Chromosomes of *S. spontaneum* clones Np-X and AP85-441 are indicated by brown and green boxes, respectively.

**Figure 6 ijms-25-06463-f006:**
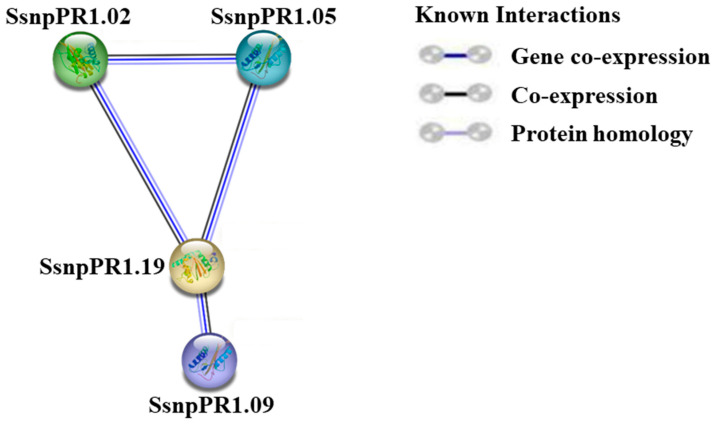
Protein–protein interactions among SsnpPR1s based on their orthologs in *Arabidopsis thaliana*. Different kinds and degrees of interaction are shown by line nodes and colors.

**Figure 7 ijms-25-06463-f007:**
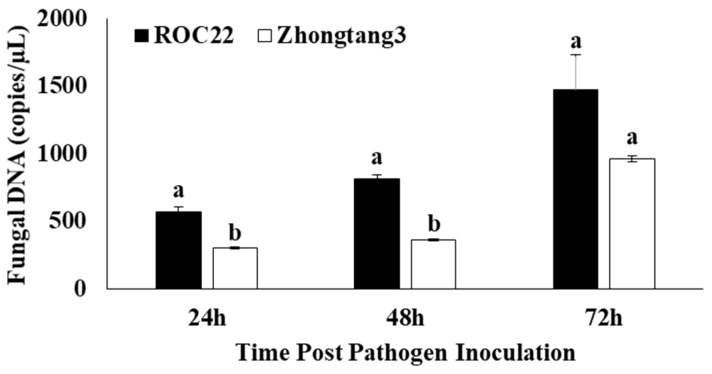
Fungal population size in two sugarcane cultivars determined by qPCR at different time points. Values are means ± standard errors. Means having the different letters above the vertical bars are significantly different at a 5% probability level.

**Figure 8 ijms-25-06463-f008:**
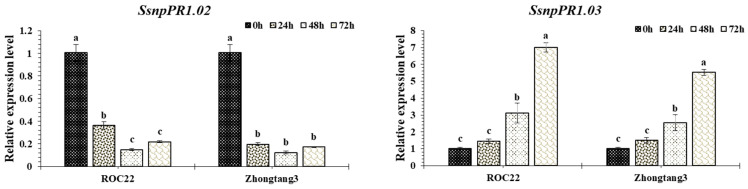
Expression profiling of *SsnPR1* genes at different time points in response to pathogen infection. Values are means ± standard errors. Different letters above bars indicates statistical differences among means.

**Table 1 ijms-25-06463-t001:** Physicochemical properties of *PR1* genes from *Saccharum spontaneum* Np-X.

Gene ID	Nomenclature	Length of Amino Acids	Molecular Weight	Isoelectric Point	Instability Index	Aliphatic Index	GRAVY
Npp.04C028570.1	SsnpPR1.01	173	18,732	7.5	58.9	69.5	−0.18
Npp.04D024970.1	SsnpPR1.02	173	18,732	7.5	58.9	69.5	−0.18
Npp.02D002590.1	SsnpPR1.03	155	16,428	6.8	40.1	59.2	−0.23
Npp.02C003580.1	SsnpPR1.04	155	16,428	6.8	40.1	59.2	−0.23
Npp.02C003610.1	SsnpPR1.05	167	17,449	6.2	26.7	76.5	−0.05
Npp.02D002640.1	SsnpPR1.06	167	17,521	6.1	26.5	77.1	−0.06
Npp.02C003670.1	SsnpPR1.07	273	28,694	8.9	38.5	76.5	−0.17
Npp.02C003650.1	SsnpPR1.08	271	27,796	8.5	35.9	79.5	−0.02
Npp.02C003630.1	SsnpPR1.09	216	23,213	8.8	56.1	82.6	−0.07
Npp.01C028770.1	SsnpPR1.11	291	31,117	10.1	54.7	66.2	−0.29
Npp.02A009930.1	SsnpPR1.12	171	18,146	6.3	34.7	66.8	−0.11
Npp.02B011150.1	SsnpPR1.13	171	18,054	7.6	30.2	71.9	0.02
Npp.02C003600.1	SsnpPR1.14	170	17,910	4.4	38.7	72.4	−0.09
Npp.10C002330.1	SsnpPR1.15	167	18,189	8.6	31.9	68.4	−0.27
Npp.10C002360.1	SsnpPR1.16	167	18,189	8.6	31.9	68.4	−0.27
Npp.04C028540.1	SsnpPR1.17	181	19,163	9.9	50.8	69.7	−0.07
Npp.04D024890.1	SsnpPR1.18	181	19,163	9.9	50.8	69.7	−0.07
Npp.04A034860.1	SsnpPR1.19	181	19,226	9.8	55.9	66.5	−0.11

**Table 2 ijms-25-06463-t002:** Evolutionary relationships among PR1 family members of *S. spontaneum* clones AP85-441 and Np-X.

Chr2B	Npp.02B011150.1.t1	==	SsChr2A	Sspon.02G0020390-1A
Chr2B	Npp.02B011150.1.t1	==	SsChr2B	Sspon.02G0020390-2B
Chr2B	Npp.02B011150.1.t1	==	SsChr2C	Sspon.02G0020390-3C
Chr2C	Npp.02C003600.1.t1	==	SsChr1A	Sspon.01G0029040-1A
Chr2C	Npp.02C003600.1.t1	==	SsChr2A	Sspon.01G0029040-1P
Chr2C	Npp.02C003610.1.t1	==	SsChr2A	Sspon.02G0024520-1A
Chr2C	Npp.02C003630.1.t1	==	SsChr2A	Sspon.02G0024500-1A
Chr2C	Npp.02C003610.1.t1	==	SsChr2B	Sspon.02G0024520-2B
Chr2C	Npp.02C003630.1.t1	==	SsChr2B	Sspon.02G0024500-1P
Chr2C	Npp.02C003610.1.t1	==	SsChr2C	Sspon.02G0024530-3C
Chr2C	Npp.02C003600.1.t1	==	SsChr2C	Sspon.01G0029040-3C
Chr4D	Npp.04D024890.1.t1	==	SsChr4B	Sspon.04G0002650-2B
Chr4D	Npp.04D024970.1.t1	==	SsChr4B	Sspon.04G0002610-2B
Chr4D	Npp.04D024890.1.t1	==	SsChr4C	Sspon.04G0002650-1P
Chr4D	Npp.04D024970.1.t1	==	SsChr8D	Sspon.04G0002610-3D
Chr10C	Npp.10C002360.1.t1	==	SsChr8A	Sspon.08G0016170-1A
Chr10C	Npp.10C002330.1.t1	==	SsChr8A	Sspon.08G0016290-1A
Chr10C	Npp.10C002330.1.t1	==	SsChr8B	Sspon.08G0016290-2B
Chr10C	Npp.10C002360.1.t1	==	SsChr8D	Sspon.08G0016170-3D
Chr10C	Npp.10C002330.1.t1	==	SsChr8D	Sspon.08G0016290-3D

**Table 3 ijms-25-06463-t003:** Primers sequences used in this study.

Gene Name	Primer/Probe Name	Forward Primer (5′→3′)	Reverse Primer (3′→5′)
*PR1.02*	q-PR1.02-F/R	GCTCAAGTACACGGAGGAGA	AGTTGGAGCCGTAGTCGTAG
*PR1.03*	q-PR1.03-F/R	ACGGTGAGAACCTCTTCTGG	GGTCGTAGTACTGCCTCTCC
*PR1.04*	q-PR1.04-F/R	CCGAGAAGCAGTTCTACGAC	AGTTGCAGGTGATGAAGACG
*PR1.05*	q-PR1.05-F/R	CTGTGGCCTTAGCAACCATC	ATACGTGGCTAGGCTCTCG
*PR1.06*	q-PR1.06-F/R	CTGTGGCCTTAGCAACCATC	AGGCTCTCATTCCACGACAC
*PR1.07*	q-PR1.07-F/R	TCGAGCTTCACAGACTGGTT	CGCCTTGTTGTGCAAGTCTA
*PR1.08*	q-PR1.08-F/R	TCGTCAACTTGCACAACGAG	AGAAGAGGTTCTCGCCGTAG
*PR1.09*	q-PR1.09-F/R	TACGACCACGCCACTAACA	GAGTCTAGTACGGGCTCTGG
*PR1.11*	q-PR1.11-F/R	CCACTATGCCTCACCCATCA	CTGAGGTGGTGCTTGCTATG
*PR1.19*	q-PR1.19-F/R	CTACGACAGGTCCACCAACT	TCAGTAGGGCCTCATCCC
*GAPDH*	GAPDH-F/R	CACGGCCACTGGAAGCA	TCCTCAGGGTTCCTGATGCC

## Data Availability

All the data supporting the findings of the current research are within this manuscript and its Appendix A.

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
