# Peer review of "Pathogenesis-Related 1 (PR1) Protein Family Genes Involved in Sugarcane Responses to *Ustilago scitaminea* Stress"

_ijms, 2024, doi:10.3390/ijms25126463_

Round 1

Reviewer 1 Report

Comments and Suggestions for Authors

Dear author,

I reviewed the manuscript entitled “Pathogenesis-related 1 (PR1) Protein Family Genes Involved in Sugarcane Responses to Ustilago scitaminea Stress”.

This research reveals identification and characterization of PR1 family genes in sugarcane (Saccharum spontaneum Np-X) and expression levels of some important genes in resistant and susceptible sugarcane cultivars (ROC22 and Zhongtang3) against the pathogen infection.

Chu et al. (2022) identified the PR1 family genes in the autopolyploid S. spontaneum AP85-441 and this research redefined PR1 family genes in sugarcane (Saccharum spontaneum Np-X). What are the differences between S. spontaneum AP85-441 and Saccharum spontaneum Np-X)? The importance of plant selection should be explained in detail in the introduction section.

Why was only the GAPDH gene used as housekeeping?

Examining the expression levels between resistant and susceptible sugarcane cultivars and handling Ustilago scitaminea made the study original.

Although the research was prepared for the purpose, more detailed information should be given about the role of the PR1 family in Sugarcane against to Ustilago scitaminea infection. The discuss should be improved.

This manuscript is written in acceptable good English.

My general opinion is that the article is of a quality that can be published after minor revision. I also believe that it will receive citations as studies in this field increase.

Sincerely

Author Response

Comment: I reviewed the manuscript entitled “Pathogenesis-related 1 (PR1) Protein Family Genes Involved in Sugarcane Responses to Ustilago scitaminea Stress”. This research reveals identification and characterization of PR1 family genes in sugarcane (Saccharum spontaneum Np-X) and expression levels of some important genes in resistant and susceptible sugarcane cultivars (ROC22 and Zhongtang3) against the pathogen infection.

Response: Esteemed reviewer, thank you for your worthy comments/suggestions for the improvement of the current manuscript. We are grateful for your valuable time to review the current manuscript.

Comment: Chu et al. (2022) identified the PR1 family genes in the autopolyploid S. spontaneum AP85-441 and this research redefined PR1 family genes in sugarcane (Saccharum spontaneum Np-X). What are the differences between S. spontaneum AP85-441 and Saccharum spontaneum Np-X)? The importance of plant selection should be explained in detail in the introduction section.

Response: Modern sugarcane cultivars (Saccharum spp. hybrid, 2n = 100–130) are highly polyploids and aneuploids derived from interspecific hybridization between S. officinarum (2n = 8x = 80) and S. spontaneum (2n = 4x–16x = 40–128) (Ali et al., 2019; Zhang et al., 2022). Moreover, S. spontaneum exhibits three basic chromosome numbers (x = 8, 9, or 10) (Zhang et al., 2022). The high variation in ploidy levels of S. spontaneum genomes contributes to the study of the evolution of autopolyploidy genomes in plants (Piperidis and D’Hont, 2020; Zhang et al., 2022). The two tetraploid genomes of S. spontaneum clones AP85-441 (1n = 4x = 32, x = 8) and Np-X (2n = 4x = 40, x = 10) have been published (Zhang et al., 2018, 2022), which provides a powerful resource to identify many gene families. Moreover, genome-wide characterization PR1 family genes in S. spontaneum clone Np-X (2n = 4x = 40, x = 10) and expression profiles in response to fungal stimuli in two contrasting sugarcane cultivars (ROC22 and Zhongtang3) remains unclear. Therefore, this study marks the systematic identification and characterization of PR1 family members in sugarcane (S. spontaneum clone Np-X), and expression profiling in two cultivars inoculated by pathogenic fungi. This study provide a solid foundation for subsequent functional characterization (reverse/forward genetics) of PR1 family members in sugarcane in response to biotic stressors.

Dear reviewer, we have also added the above mentioned logics behind the selection of S. spontaneum clone Np-X (2n = 4x = 40, x = 10) in introduction section of the manuscript.

References:

Ali, A., Khan, M., Sharif, R., Mujtaba, M. and Gao, S.J., 2019. Sugarcane omics: an update on the current status of research and crop improvement. Plants, 8(9), p.344.

Zhang, Q., Qi, Y., Pan, H., Tang, H., Wang, G., Hua, X., Wang, Y., Lin, L., Li, Z., Li, Y. and Yu, F., 2022. Genomic insights into the recent chromosome reduction of autopolyploid sugarcane Saccharum spontaneum. Nature Genetics, 54(6), pp.885-896.

Piperidis, N. and D’Hont, A., 2020. Sugarcane genome architecture decrypted with chromosome‐specific oligo probes. The Plant Journal, 103(6), pp.2039-2051.

Zhang, J., Zhang, X., Tang, H., Zhang, Q., Hua, X., Ma, X., Zhu, F., Jones, T., Zhu, X., Bowers, J. and Wai, C.M., 2018. Al-lele-defined genome of the autopolyploid sugarcane Saccharum spontaneum L. Nature genetics, 50(11), pp.1565-1573.

Comment: Why was only the GAPDH gene used as housekeeping?

Response: Since the GAPDH expression level is usually not affected under experimental or physiological conditions as evidenced from previous literature, so GAPDH was only used as an internal control in this study.

Comment: Examining the expression levels between resistant and susceptible sugarcane cultivars and handling Ustilago scitaminea made the study original. Although the research was prepared for the purpose, more detailed information should be given about the role of the PR1 family in Sugarcane against to Ustilago scitaminea infection. The discuss should be improved.

Response: We have improved the discussion section with logical reasoning as per your suggestions. We believe that the corrections will meet with the approval.

Comment: This manuscript is written in acceptable good English. My general opinion is that the article is of a quality that can be published after minor revision. I also believe that it will receive citations as studies in this field increase.

Response: Thank you for your appreciated comments.

Reviewer 2 Report

Comments and Suggestions for Authors

In this study, Javed et al. report to annotate eighteen PR1 members in Saccharum spontaneum Np-X genome. They analyzed the structure, cis-element, duplication for each member based on bioinformatic analysis. Also, they performed the transcription analysis of PR1s in two sugarcane cultivars in response to a fungal pathogen of Ustilago scitaminea. However, currently results are preliminary to support some new findings.

Major concerns:

1. I would suggest that the author should validated the predicted interactions among on SsnpPR1.02, SsnpPR1.05 and SsnpPR1.19.

2. Because the pathogenesis-related proteins are ordinarily defined to be induced expression after pathogens inoculation, and to be directly inhibited the growth of pathogens. So I suggest the author to test the inhibition ability of Ustilago scitaminea for each members. 

Minor concerns:

Line 42-44, PR should be full name as the first coming out. And how PR genes control the expression of other proteins?

Line 45, I suggest the author to define the PR protein: pathogen inducible expression, growth inhibition to microbe, etc?

Line 49-50, nuclear magnetic resonance (NMR) technology, and latin names should be italicized.

Line 54-55, Line 60-61, two sentences duplicated.

Line 84, give a small introduction how to identified 18 and 17 PRs from Np-X and AP85-441, respectively.

 Line 86-87, Notably, the putative SsnpPR1 proteins contains 155 to 291 amino acids.....

Line 102-106, whereas other 14 SsnpPR1s are intron less. And provide the universal names for each SsnpPR1 members, as a example, there are three types as SsnpPR1.7/1.07/1-7.

The format of references and citations need to be corrected based on journal's required

Comments on the Quality of English Language

need to be polish.

Author Response

Esteemed reviewer,

Please find as attached herewith the response file for your worthy comments/suggestions. 

Reviewer 3 Report

Comments and Suggestions for Authors

Javed et al. presented Pathogenesis-related 1 (PR1) Protein Family Genes Involved in Sugarcane Responses to Ustilago scitaminea Stress.

The manuscript is overall well presented. Here are a couple of points authors may want to address:

Lines 54-55: more background of CAPE is needed. What are the mechanisms these motif works to trigger immunity?

Line 75-76: authors did not introduce why studying However, no reports are available about the PR1 family 75 genes response to U. scitaminea infection in sugarcane, this pathosystem in particular.

Line 84-90: how these data compare to other well-known PR1 proteins (e.g. Arabidopsis and Solanum lycopersicumPR1). Authors may include them in the phylogenetics analysis. Authors may also want to do an ML tree, in addition to using NJ.

Line 117-119: did the whole genome duplication happen may explain these segmental duplications?

The quality of Figure 4 needs to be improved: the numbers in the plots are collapsed.

Author Response

Javed et al. presented Pathogenesis-related 1 (PR1) Protein Family Genes Involved in Sugarcane Responses to Ustilago scitaminea Stress.

The manuscript is overall well presented.

Response: Esteemed reviewer, thank you for your worthy and valuable comments/suggestions for the improvement of the current manuscript.

Here are a couple of points authors may want to address:

Comment: Lines 54-55: more background of CAPE is needed. What are the mechanisms these motif works to trigger immunity?

Response: Dear reviewer, thank you for your comment. We have added the more background about CAPE and brief mechanism of CAPE induced regulation of PR-1 expression to trigger immunity. However detailed information about this mechanism can be accessed through our recently published manuscript dealing with pathogenesis related-1 proteins in plant defense. We have also cited it in text for your reference.

  1. Javed, T., Wang, W., Yang, B., Shen, L., Sun, T., Gao, S.J. and Zhang, S., 2024. Pathogenesis related-1 proteins in plant defense: regulation and functional diversity. Critical Reviews in Biotechnology, pp.1-9.

Comment: Line 75-76: authors did not introduce why studying However, no reports are available about the PR1 family 75 genes response to U. scitaminea infection in sugarcane, this pathosystem in particular.

Response: We have included the reason/logic as per your suggestion.

Sugarcane (Saccharum spp.) is a major dual-purpose cash and biofuel crop worldwide [15]. The highly polyploid and aneuploidy modern sugarcane cultivars (Saccharum spp. hybrid, 2n = 100–130) are derived from interspecific hybridization between S. officinarum (2n = 8x = 80) and S. spontaneum (2n = 4x–16x = 40–128) [16,17]. Additionally, the increased variations in S. spontaneum genomes polyploidy levels contributes to the study of the evolution of autopolyploidy genomes in plants [17,18]. Importantly, the two reference S. spontaneum clones Np-X (2n = 4x = 40, x = 10) and AP85-441 (1n = 4x = 32, x = 8) tetraploid genomes offers a valuable resource to uncover numerous gene families with functional diversity [17,19]. Previously, PR1 family genes were identified in the autopolyploid S. spontaneum AP85-441 and their temporal transcript patterns were examined in response to different stressors [5]. However, no reports are available about PR1 family genes response to U. scitaminea infection in sugarcane. Therefore, this study marks the systematic identification and characterization of PR1 family members in sugarcane (S. spontaneum clone Np-X), and expression profiling in two cultivars inoculated by pathogenic fungi. This study provide a solid foundation for subsequent functional characterization (reverse/forward genetics) of PR1 family members in sugarcane in response to biotic stressors.

Comment: Line 84-90: how these data compare to other well-known PR1 proteins (e.g. Arabidopsis and Solanum lycopersicumPR1). Authors may include them in the phylogenetics analysis. Authors may also want to do an ML tree, in addition to using NJ.

Response: Thank you for the suggestion. We have included the PGT including PR1 proteins from Saccharum spontaneum clones AP85-441 and Np-X in main text file. Based on phylogeny, all PR1 genes were clustered into four groups. Group A comprised of 11 and 12 SsPR1 and SsnpPR1 genes, respectively. Additionally, 2 different genes from each clone were clustered in groups B and D separately. Group C comprised three (SsPR1.11/15/16) AP85-441 genes and three genes (SsnpPR1.11/15/16) from Np-X. We think that addition of another phylogeny with different crops from different crop categories (monocot and dicots) will have less significance. Moreover, we have tried both methods but phylogeny based on NJ method was much more accurate than an ML tree based on an alignment of PR1s from S. spontaneum clones AP85-441 and Np-X. Therefore, we have added PGT based on NJ method.

Comment: Line 117-119: did the whole genome duplication happen may explain these segmental duplications?

Response: Yes you are right polyploidy or whole-genome duplication (WGD) is a major event that drastically reshapes genome architecture and is often assumed to be causally associated with organismal innovations and radiations. The quality of Figure 4 needs to be improved: the numbers in the plots are collapsed. We have discussed both polyploidy and duplication events in discussion section of this manuscript.

Comment: The quality of Figure 4 needs to be improved: the numbers in the plots are collapsed.

Response: We have improved the quality of the figure and added a high pixels (600 dpi) figure in the manuscript file as per your suggestion.